# Redox Homeostasis and Metabolic Profile in Young Female Basketball Players during in-Season Training

**DOI:** 10.3390/healthcare9040368

**Published:** 2021-03-25

**Authors:** Rosamaria Militello, Simone Luti, Matteo Parri, Riccardo Marzocchini, Riccardo Soldaini, Alessandra Modesti, Pietro Amedeo Modesti

**Affiliations:** 1Department of Biomedical, Experimental and Clinical Sciences “Mario Serio”, University of Florence, 50134 Florence, Italy; rosamaria.militello@unifi.it (R.M.); simone.luti@unifi.it (S.L.); matteo.parri@unifi.it (M.P.); riccardo.marzocchini@unifi.it (R.M.); 2Department of Experimental and Clinical Medicine, University of Florence, 50134 Florence, Italy; riccardo.soldaini@gmail.com (R.S.); pamodesti@unifi.it (P.A.M.); 3Viale G. Morgagni, 50, 50134 Firenze, Italy

**Keywords:** exercise training, oxidative stress, adiponectin, metabolomics, female athletes

## Abstract

Background: Most studies on oxidative stress markers and antioxidant levels have been conducted in male athletes, although female participation in sport has increased rapidly in the past few decades. In particular, it could be important to assess oxidative stress markers in relation to the training load because the anaerobic path becomes predominant in high-intensity actions. Methods: Ten female professional basketball players, performing five 2 h-lasting training sessions per week, and 10 sedentary control women were investigated. Capillary blood and saliva samples were collected in the morning before the training session. The antioxidant capacity and the levels of reactive oxygen metabolites on plasma were determined measuring Reactive Oxygen Metabolite and Biological Antioxidant Potential (d-ROMs and the BAP Test). Salivary cortisol was detected by using commercial enzyme-linked immunosorbent assay kit. Results: The antioxidant capacity (BAP value) was significantly higher in elite basketball players (21.2%; *p* < 0.05). Conversely, cortisol (51%; *p* < 0.009) and the levels of oxidative species (d-ROM, 21.9%; *p* < 0.05) showed a significant decrease in elite athletes.

## 1. Introduction

In elite athletes, there is large disparity among different training protocols and the effects on oxidative stress and inflammatory cytokines are still not well known. In a previous review, we concluded that chronic training seems to be responsible for some perturbations in different metabolic hormones linked to inflammation and to plasma redox homeostasis [1]. Therefore, the correct training program is essential. Understanding the alterations of redox homeostasis in well-trained athletes will help to clarify the complex mechanism of adaptation to acute exercise and chronic training and to use redox signals as possible markers of training stress [2]. Moreover, the exact regulatory roles of redox signals will help to understand mechanisms to improve exercise performance and prevent overreaching/overtraining, a phenomenon in which there is an increase in pro-inflammatory markers and consequently a decrease in sport performance [2]. Most of the studies on changes in oxidative stress markers and antioxidant levels have been conducted in male athletes, but it is possible to find differences in antioxidant and inflammatory responses among male and female athletes, since among men and women, different hormonal status exists as well as different ratio of progesterone/estradiol. In recent years, women’s participation in basketball, a very popular sport played worldwide, has rapidly increased [3]. Basketball is a sport characterized by aerobic and anaerobic intermittent high-intensity demands [4]. However, the anaerobic pathway becomes preponderant during high-intensity actions (such as accelerations, decelerations, changes of direction, jumps, sprints, contacts) [5]. Until today, there are few data in elite female basketball athletes regarding both metabolic and oxidative stress modifications as well as the release of peripheral molecules. It is known that in skeletal muscle, differences regarding muscle fibers’ type distribution as well as substrate availability are highly relevant in men and women [6]. Moreover, the identification of estradiol receptors in skeletal muscle has opened studies on a role of 17-β estradiol in regulation of substrate metabolism, the most important female sex hormone [7]. In women, compared to men, at the same exercise intensity, the overall energy expenditure is to the advantage of the fat oxidation promoted by higher circulating estrogens, and women’s skeletal muscle fibers have increased maximal capacity to oxidize fatty acids and lactate [8]. Prolonged intense physical activity could increase muscle damage that, in professional athletes, could lead to perturbations in several metabolic hormones related to inflammation. Adiponectin is a signaling hormone with an anti-inflammatory effect and its plasma level is modified by training [1]. Several studies conducted in women reported a high level of plasma adiponectin compared with men, as well as a positive correlation seen between circulating adiponectin and whole-body insulin sensitivity in women [9]. However, in women, there is a lower expression of adiponectin receptor 1 in skeletal muscle when compared to men, and this could limit the effects of the high serum level of adiponectin [10]. Therefore, it will be interesting to deepen the investigation into whether the sex diversity in adiponectin has direct metabolic effects in skeletal muscle metabolism. Since cortisol, a stress hormone, plays crucial control roles in the body, reducing inflammatory processes and increasing blood sugar, it becomes important to evaluate its variation in subjects who practice sports at high levels [11]. In women, cortisol level shows an increase in response to an acute bout of endurance exercise, but it appears to decline with endurance training [12]. Moreover, resistance training has little effect on resting hormone levels. An elevated resting cortisol level has been associated with overtraining in athletes [13]. In conclusion, in women, it is difficult to conclude unambiguous cortisol responses to exercise and training. The aim of this study is to determine the plasma level together with the metabolic profile and redox homeostasis in elite female athletes who practice basketball at a competitive level in comparison with healthy female controls.

## 2. Materials and Methods

### 2.1. Materials

Unless specified, all reagents were obtained from Sigma (St. Louis, MO, USA), except Polyvinylidene Fluoride (PVDF) membrane (Millipore, Bedford, MA, USA). 

### 2.2. Participants 

A cohort of 10 female professional basketball players and 10 women who do not practice any sports were involved in this study. The elite-level female athletes were recruited from the local sports clubs in Florence Italy “US AFFRICO-Firenze” and have been practicing this sport for more than 5 years. The athletes trained at least 5 times a week according to specific training programs, with sessions lasting 2 h per day. The training protocol involved technical and aerobic exercise. Team coaches tracked day-to-day training data for each player and for each training session over the entire season. Team coaches followed a classical training program that we can resume as follows: half an hour of low–moderate run followed by interval training runs with sprinting and repeating to improve speed and simulate the different game situations. Then, move on to strengthening the muscles involved in basketball: (A) work on the shoulder muscles by performing the classic front and side lifts with weight-lifting dumbbells: three sets of twelve repetitions. (B) work with the abdominals (oblique and the rectus abdominis muscle) which are essential to maintain balance in every movement and to contribute to a good deadlift in jumps: three sets of 15 repetitions. In the present study, the athletic trainers used the rating of perceived exertion (RPE) system to monitor training load (TL) to know the individual responses of athletes in different training sessions [14]. 

The control female group was recruited among students of the degree course in Motor Sciences, Sport and Health of the University of Florence. All female subjects who volunteered to participate following explanation of all experimental procedures were adults (≥20 years) and of Caucasian ethnicity. A medical history and physical activity questionnaire were completed by the participants in order to determine the eligibility. None of them used antioxidant or nutritional supplements. Subjects were selected because of non-smoking status, age and stable body weight. All women were enrolled randomly with respect to menstrual cycle. After initial screening, all participants received a complete explanation of the purpose, risks and procedures of the study. A written informed consent form was provided prior to enrolment in the study, it was conducted according to the policy statement set forth in the Declaration of Helsinki, and all the experiments were conducted according to established ethical guidelines. The study was approved by the local Ethics Committee of the University of Florence, Italy (AM_Gsport 15840/CAM_BIO). All measurements are performed in resting conditions and by the same operator. During the study period, the trained subjects had their optimal body composition (i.e., lowest fat mass and highest fat-free mass). Weight is measured to the nearest 0.1 kg and height (H) to the nearest 0.5 cm. Body mass index (BMI) was calculated from the ratio of body weight (kg) to body height (m^2^). A capillary blood sample was taken using a heparinized Microvette CB300 (Sarstedt AG and Co, Nümbrecht, Germany) from each volunteer after obtaining written and informed consent. Holding the Microvette in a horizontal position, the blood sample was collected with the capillary tip (filling volume 300 µL). The capillary collection was preferred to the venous one because of its reduced invasiveness and simple execution; moreover, small volumes of samples were sufficient for the experiments carried out. All the biological samples were collected in the morning and as regards the athletes, the samples were collected away from sport competitions and before the daily training sessions.

### 2.3. Saliva Collection and Cortisol Measurement

Saliva samples of each participant were collected using salivette Cortisol (Sarstedt AG and Co) at the beginning of each training session, and to avoid any variations due to circadian rhythm, all saliva samples were taken at the same time of day. Participants were instructed to not eat, not drink and not brush teeth 30 min before the collection. The cotton sliver of the salivette was taken out and put in the sublingual for 1/2 min, and then put back to the salivette. The samples were centrifuged at 2000 rpm for 2 min. Saliva flow was collected from salivette’s bottom and was stored at −20° for further laboratory analysis.

Salivary cortisol was detected using commercial enzyme-linked immunosorbent assay kit for cortisol (Item No.500360 Cayman hem): 50 µL of saliva samples not diluted were used and the assay was performed according to manufacturer’s recommended procedures. The sensitivity provided by the manufacturer is approximately 35 pg/mL with a detection range from 6.6 to 4000 pg/mL. The samples were analyzed in duplicate [11].

### 2.4. Plasma Oxidative Stress Measurements 

The antioxidant capacity and the levels of reactive oxygen metabolites on plasma were determined by using the d-ROM test (diacron-Reactive Oxygen Metabolite) and the BAP Test (Biological Antioxidant Potential). The biomarkers d-ROM and BAP were selected based on their long-term stability. The measurements were carried out according to the manufacturer’s instructions. Briefly, the d-ROM test reflects the amount of organic hydroperoxides that is related to the free radicals from which they are formed. When the samples are dissolved in an acidic buffer, the proteins in the acidic medium liberate the metal ions (Fe) and the hydroperoxides react with them and are converted to alkoxy and peroxy radicals. These newly formed radicals oxidize an additive aromatic amine (N,N-diethylparaphenylen-diamine) and cause formation of a relatively stable colored cation radical that is spectrophotometrically detectable at 505 nm. The BAP test provides an estimate of the global antioxidant capacity of the sample, measured as its reducing potential against ferric ions. The intensity of the discoloration resulting by mixing a ferric chloride solution with a thiocyanate derivative solution is spectrophotometrically detectable at 505 nm and is proportional to the ability of plasma or serum to reduce ferric ions. Analyses were performed using a free radical analyzer system (FREE Carpe Diem, DIACRON INTERNATIONAL s.r.l) that included a spectrophotometric device reader and a thermostatically regulated minicentrifuge. The d-ROM results are expressed in arbitrary units (U. Carr), one unit of which corresponds to 0.8 mg/L of hydrogen peroxide. The BAP results are expressed in µmol/L of the reduced ferric ions [15].

### 2.5. Adiponectin Western Blot Analysis

Plasma samples were clarified by centrifugation and total protein contents were obtained using the Bradford assay. An equal amount of each sample (12.5 μg of total proteins) was added to 4× Laemmli buffer (0.5 M TrisHCl pH 6.8, 10% Sodium dodecyl sulfate (SDS), 20% glycerol, β-mercaptoethanol, 0.1% bromophenol blue) and boiled for 5 min. Samples were separated on 12% SDS/ polyacrylamide gel electrophoresis (PAGE) and transferred onto PVDF membrane using Trans-Blot Turbo Transfer System (BIO-RAD). PVDF was probed with primary antibody (Acrp30 Santa Cruz) diluted 1:1000 in 2% milk and incubated overnight at 4 °C. After incubation with horseradish peroxidase (HRP)-conjugated anti-mouse IgG (1:10,000) (Santa Cruz Laboratories), immune complexes were detected with the enhanced chemiluminescence (ECL) detection system (GE Healthcare) and by an Amersham Imager 600 (GE Healthcare). For quantification, the blot was subjected to densitometric analysis using the ImageJ program. The intensity of the immuno-stained bands was normalized with the total protein intensities measured by Coomassie brilliant blue R-250 from the same PVDF membrane blot as previously reported [16].

### 2.6. Gas Chromatography—Mass Spectrometry (GC-MS) Analysis of Plasma Samples

Metabolite analysis of plasma samples of basketball players and control women were performed as reported by Luti [16]. Briefly, 100 µL of plasma samples were subjected to methanol/chloroform precipitation and the upper phase was evaporated at room temperature in a rotovapor. Obtained metabolites were analyzed by GC-MS technique after their derivatization with N-Trimethylsilyl-N-methyl trifluoroacetamide (MSTFA). The MassHunter data processing tool (Agilent) was used to obtain global metabolic profiling using the Fihen Metabolomics RTL library (Agilent G1676AA). Statistical analysis was performed by GraphPad Prism 8.0 (GraphPad Software Inc., La Jolla, CA, USA).

### 2.7. Statistical Analysis 

Data are presented as means ± standard deviation (SD) from at least three experiments. Statistical analysis was performed by two-tailed t-test using Graphpad Prism 8. Significance was defined as *p* < 0.05. For Western blot quantification, the blot was subjected to densitometric analysis using the ImageJ program. For correlation analysis of adiponectin level with each parameter measured, non-parametric correlation Spearman test was performed using Graphpad Prism 8. A 90% confidence interval as a range of values was used and *p* < 0.1 was considered as significant.

## 3. Results

### 3.1. Participants’ Characteristics

The female control students were age/BMI-matched with the athletes and descriptive characteristics of the participants are reported in Table 1, showing that mean age was 26 ± 4.13 years for all volunteers (athletes and non-athletes). For female basketball players, mean weight was 68.7 ± 11.9 kg and height was 175.6 ± 8.3 cm; for female controls, mean weight was 58.7 ± 5.8 kg and height was 163.4 ± 5.9 cm. Basketball players were heavier and taller than controls. However, no significant differences were found regarding BMI between groups (22.07 ± 2.12 for all). All the enrolled participants did not have metabolic disorders and did not take specific drugs, as reported in the Methods Section. The control subjects were sedentary individuals and as reported in the Methods Section, all women were enrolled randomly with respect to menstrual cycle. 

### 3.2. Plasma Oxidative Stress Measurements 

We measured the antioxidant capacity (BAP) and the levels of oxidative species (d-ROM) in plasma from all the participants and the results obtained are reported in Figure 1. We noticed that comparing BAP and d-ROM (Figure 1A,B), significant differences exist between the two groups. In detail, analyzing the antioxidant capacity (BAP value), a significant increase (21.2%; *p* < 0.05) in basketball players is observed compared to the control group; as shown in Figure 1A, the BAP mean value was 1774.73 ± 169 µmol/L for basketball players in comparison with 1465.04 ± 139 µmol/L for controls. Moreover, as reported in Figure 1B, the basketball players show a decrease of 21.9% (*p* < 0.05) of the levels of oxidative species (d-ROM), the mean value for athletes was 280.5 ± 38 UCarr and for the control group was 359.75 ± 44 UCarr.

### 3.3. Plasma Adiponectin Determination

The plasma adiponectin level was determined by Western blot in both groups and the results reported no differences between players and controls (Figure 1C). The statistical analysis performed demonstrated no significant difference between the two groups.

### 3.4. Salivary Cortisol Determination

The ELISA assay, used for determining the concentration of salivary cortisol, revealed a variation between basketball players and controls. The results, reported in detail in Figure 1D, show a significant decrease (51%; *p* < 0.009) in basketball players in comparison with the control group. In detail, the cortisol mean value was 915.09 ± 613.69 pg/mL for basketball players and 1867.68 ± 659.83 pg/mL for controls. 

### 3.5. Plasma Metabolic Profile

We characterized the metabolic profile of plasma from basketball players and control females through GC-MS, and the identification of each compound was obtained through the Fiehn library. Our method allowed us to detect a total of 39 compounds presented in the two groups, and among them, we identified 6 compounds showing a statistically different concentration (Figure 2 and Table 2). In plasma from athletes, we found an increase in lactic acid (460%; *p* = 0.0013), urea (220%; *p* = 0.0361) and ornithine (130%; *p* = 0.0467), and a reduction in hydroxybutyric acid (75%; *p* = 0.0258), L-glutamic acid (83%; *p* = 0.0353) and L-asparagine (82%; *p* = 0.0498). We determined the Spearman’s correlation among adiponectin, BAP and d-ROM with the metabolites that showed variation between the two groups. The only correlation was shown between biological antioxidant potential and urea (r = 0.83; *p*-value = 0.06) in female athletes (data not shown). 

### 3.6. Metabolomic Interaction Network

We built an interaction network using MetScape (http://metscape.med.umich.edu (accessed on 10 December 2020) from Cytoscape) which provides a bioinformatics tool for the interpretation of metabolomic data [17]. We used the metabolites obtained from plasma analysis to trace the connections between them and genes and visualize compound networks using a database that integrates data from Kyoto Encyclopedia of Genes and Genomes (KEGG) and EHMN (Edinburgh Human Metabolic Network). We built the input list using the metabolites that showed a significant increase/decrease in plasma from basketball players in comparison with controls (Table 2). Figure 3 reports the selected networks obtained. The involvement of butanoate and folate metabolism (Figure 3A,B) as well as glycolysis and gluconeogenesis (Figure 3C) is evident. The urea cycle and the metabolism of several amino acids such as arginine, proline, glutamate, aspartate and asparagine (Figure 3F) are also involved. Moreover, we noted an involvement of purine and histidine metabolism (Figure 3D,E) 

## 4. Discussion

### 4.1. Redox Profile

The purpose of this study was to identify a metabolic and redox profile on female basketball players during the seasonal training period. Moreover, this investigation aimed to be descriptive and exploratory in an attempt to deepen the understanding of the biochemical alterations that occur during the monitoring of female athletes. However, only a limited number of studies have focused on oxidative stress in female athletes. Dopsaj [18] showed that female athletes were more prone to oxidative stress than males. Arsic [19] investigates the activities of antioxidant enzymes in plasma and erythrocytes in well-trained female athletes practicing anaerobic/aerobic sports. They demonstrated that prolonged strenuous physical activity increases lipid peroxidation (LPE), total antioxidant status, H2O2 and oxidized glutathione (GSSG) levels and the catalase (CAT) activity in female athletes. Moreover, they conclude that these results could be explained by adaptation mechanisms of antioxidative defense which depends on the type of sport. The results of a study by Souglis [20] report that after a soccer match, sex and playing position show different effects on oxidative stress, inflammation molecules and muscle damage. The authors reported that for the same player position, there are higher values in male players compared to female players. Moreover, estrogens play a protective role in the process of exercise-induced muscle damage by maintaining cellular membrane stability. The 17β-estradiol might reduce the inflammation due to the intense eccentric physical activity. We therefore studied oxidative stress in basketball as this parameter is still poorly evaluated in female athletes. In basketball, the intermittent high-intensity activity during matches requires a high function of both aerobic and anaerobic energy systems, although the anaerobic component is preponderant [4]. In our experimental conditions in female elite athletes with body mass index comparable to that of the reference controls, the antioxidant capacity showed a significant increase, and as expected, a reduction in oxidative species. 

### 4.2. Adiponectin 

Adiponectin secreted and expressed by skeletal muscle is considered a “good adipokine” due to its anti-inflammatory effect. However, there are little data on the relationship between muscular fitness and adiponectin levels as well as on associations between adiponectin and training in female athletes. Therefore, one of the aims of this study was to evaluate the level of adiponectin in the plasma of athletes in comparison to healthy young women. Morrison [21] reported recently that levels of adiponectin are known to increase with chronic exercise when associated with weight loss and reduction in body fat. According to several authors, the modification of adipokine plasma levels after a long-term moderate aerobic exercise or after intensive exercise training are associated with a weight loss and a negative energy balance [22]. Nevertheless, Plinta [23] found no changes in plasma adiponectin plasma levels in well-trained young female basketball players after short-term moderate aerobic exercise, but the authors observed an increase in adiponectin level induced by intensive fitness and speed exercise, suggesting an increase in lipolysis induced by catecholamine. They concluded that the duration and intensity of exercise are important factors influencing catecholamine release, which can potentially determine the changes in circulating adipokine levels. In our study, we observed no change in adiponectin signal intensity in female athletes in comparison to control. Hornsby [24] found an increase in plasma adiponectin level in concert with decreases in salivary cortisol in female athletes following a training periodization model, although the cytokine changes were relatively small. In our study, the female subjects’ menstrual cycles were not monitored, and Wyskida [25] reported that adiponectin is not affected by changes in menstrual cycle. 

### 4.3. Salivary Cortisol

In our study, we found a reduction of cortisol in saliva of athletes in comparison to control women, and this is in line with the results obtained by Casanova, who reports that salivary level of cortisol decreases after a match in female football players, suggesting that high-level athletes are well-adapted to competition stress because they train regularly [26]. Moreover, in our results, it appears that in female athletes, the training program induces a significant reduction in the level of salivary cortisol. The reduced salivary cortisol level suggests that their training program does not cause stress, and this could also be confirmed by the non-variation in plasma adiponectin (an anti-inflammatory cytokine) levels which may be due to a low oxidative stress owing to continuous and regular training. 

### 4.4. Metabolomic Profile

The metabolomic analysis performed in our study showed the involvement of butyrate metabolism that describes the metabolic fate of a number of short-chain fatty acids involved in the production of ketone bodies, of short-chain lipids or as precursors to the citrate cycle, glycolysis or glutamate synthesis, playing a role in glucose metabolism and lipid metabolism. Moreover, many studies have investigated the role of butyrate in inflammation, in particular it produces an anti-inflammatory effect in several tissues [27]. However, to date, and as far as we know, the only data reported on sport and butyrate is a recent study by Pintus, in which urinary butyric acid concentration is associated with the type of physical exercise [28]. We identified an increase in urea and lactic acid levels in plasma from athletes. As we previously reported, urea concentration is a useful biomarker to evaluate athletes’ level of adaptation to training, and high concentrations of urea were found in plasma from high-level athletes; moreover, a low concentration indicates the need of an increase in the levels of the exercise load [29]. Therefore, this value can be used to measure the sport performance. Moreover, in our experimental conditions, we found that female basketball players have greater lactic acid concentration, suggesting an improvement in performance, as recently reported by Hoffmann [30]. 

## 5. Conclusions

Despite that the main limitation of our study is the low number of participants analyzed, these data indicated that trained women have greater antioxidant capacity and a reduction in oxidative species than sedentary women, differences that are associated with an increase in plasma lactate level. These data complement recent observations suggesting the participation of lactate in the antioxidant potential through an intracellular buffering action that would allow it to neutralize or delay the creation of an acidic environment in muscle cells [31]. The power of these actions seems to be enhanced with training in women. Moreover, lactate metabolism fulfills other functions: as an energy source, a gluconeogenic precursor and a regulatory molecule, a “lactormone” [32]. Lactate produced after high cell work is secreted into the interstitial and circulation, and in diverse tissues, it acts by cell redox regulation, reactive oxygen species (ROS) generation and lactylation of histones. Moreover, regular exercise results in repeated lactate exposure, leading the adaptive processes such as mitochondrial biogenesis and other health characteristics, such as improved metabolic flexibility. For this reason, the fact that in our experimental conditions we found high levels of lactate, low levels of adiponectin as well as low salivary cortisol can lead to the conclusion that the type of training in basketball induces adaptation in well-trained athletes without causing inflammatory phenomena and avoids the onset of overtraining/overreaching.

## Figures and Tables

**Figure 1 healthcare-09-00368-f001:**
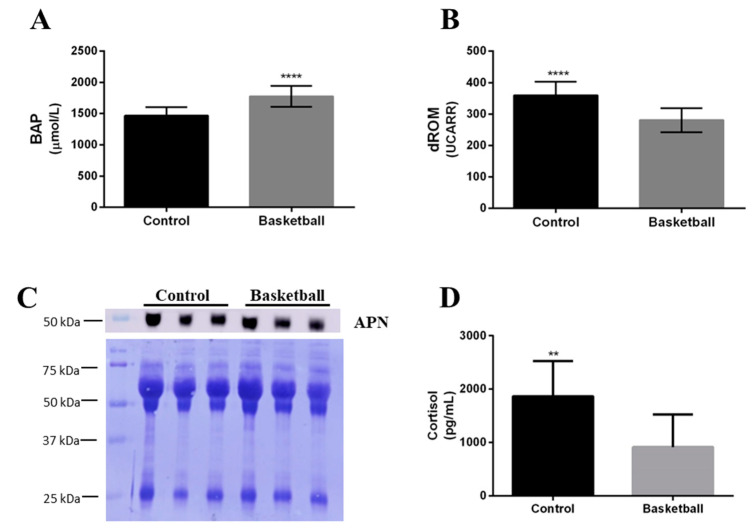
Plasma oxidative stress, adiponectin and cortisol determination. (**A**) The antioxidant capacity was evaluated using the BAP Test (Biological Antioxidant Potential), and (**B**) the levels of reactive oxygen metabolites using the d-ROM test (diacron-Reactive Oxygen Metabolite) by a free radical analyzer system (FREE Carpe Diem, DIACRON INTERNATIONAL s.r.l). (**C**) A representative immunoblot of adiponectin is shown together with the corresponding Coomassie-stained PVDF membrane. (**D**) Cortisol levels measured using enzyme-linked immunosorbent assay kit. All the measurements (*n* = 10 Control and 10 basketball) were performed in triplicate and are reported in the histograms as mean ± SD. The statistical analysis was carried out by two-tailed *t*-test using Graphpad Prism 8 (** *p* < 0.01; **** *p* < 0.0001).

**Figure 2 healthcare-09-00368-f002:**
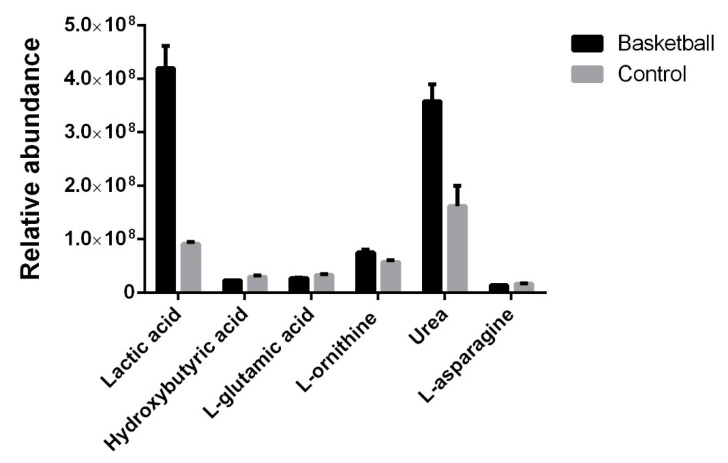
Plasma metabolomic profile using gas chromatography–mass spectrometry (GC-MS). Histogram representation of plasma metabolites whose relative abundance is statistically different between basketball athletes and controls (*p* value < 0.05).

**Figure 3 healthcare-09-00368-f003:**
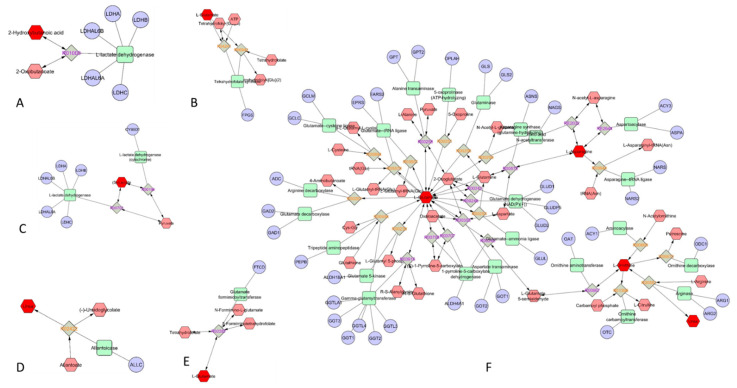
Selected interaction networks obtained using plasma metabolites identified by GC-MS showing a statistically significant increase/decrease in basketball players in comparison with controls (see Table 2). The network analysis was carried out by using the MetScape 3 App for Cytoscape (http://metscape.med.umich.edu accessed on 10 December 2020). (**A**) Butanoate metabolism, (**B**) vitamin B9 (folate) metabolism, (**C**) glycolysis and gluconeogenesis, (**D**) purine metabolism, (**E**) histidine metabolism, (**F**) urea cycle and metabolism of arginine, proline, glutamate, aspartate and asparagine.

**Table 1 healthcare-09-00368-t001:** Participants’ characteristics.

Characteristics	Mean (SD)			
	All the Study Participants	Basketball	Control	*p*
Age (year)	26 ± 4.13	25.13 ± 5.46	26.88 ± 2.32	0.4157
Weight (kg)	63.7 ± 10.4	68.7 ± 11.9	58.7 ± 5.8	0.0509
Height (cm)	169.5 ± 9.4	175.6 ± 8.3	163.4 ± 5.9	0.0043 **
BMI (kg/m^2^)	22.07 ± 2.12	22.12 ± 2.05	22.03 ± 2.32	0.9357

** Statistically significant difference *p* < 0.01. SD = standard deviation, BMI = body mass index.

**Table 2 healthcare-09-00368-t002:** List of metabolites with significant differences between female basketball players and controls, identified by gas chromatography–mass spectrometry (GC-MS) analysis.

Metabolite Number	Metabolite Name	Retention Time	CAS Number *	KEGG ID °
1	2-hydroxybutyric acid	(7.852)	565-70-8	C05984
2	L-lactic acid	(6.851)	79-33-4	C00186
3	L-asparagine	(14.984)	70-47-3	C00152
4	L-glutamic acid	(14.398)	56-86-0	C00025
5	L-ornithine	(16.632)	70-26-8	C00077
6	Urea	(9.599)	57-13-6	C00086

* Chemical Abstract Service number, ° KEGG identifier (https://www.genome.jp/kegg/ accessed on 5 November 2020).

## Data Availability

The data presented in this study are available on request from the corresponding author. The data are not publicly available due to restrictions for privacy.

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
