# Peer review of "Redox Homeostasis and Metabolic Profile in Young Female Basketball Players during in-Season Training"

_healthcare, 2021, doi:10.3390/healthcare9040368_

Round 1

Reviewer 1 Report

In this manuscript the authors provide insight into the female athlete body’s response to exercise training, with respect to stress.  The authors use saliva to measure cortisol, and plasma to measure oxidants, metabolites, and adiponectin expression. 

Suggestions to authors:

(1) Treat your control, as your control or "normal" to make your results more readable. Accordingly, report your data in reference to your control not your experimental group; for example, "basketball players had 51% less XXX than control."  NOT, "control had 51% more XXX compared to basketball."

(2) Authors refer to the basketball player subjects as “professional” or “elite” athletes but don’t provide other information to further define or support these terms other than that they trained 5 times a week for 2h per session.  Please further define these parameters with regards to your introduction about chronic or acute training.  For example, how many months or years have they been doing this?

(3) Grammar and spelling throughout the manuscript need to be addressed.

(4) Authors don’t give any information about how and where they recruited the Control group.  They also don’t give any details of what they are trying to “match” for the Control characteristics.  For example, are they doing an age-matched control group?  Ethnicity-matched control group?  BMI-matched control group?  Please explain.   Authors also don’t give any information regarding ethnicity of any of the participants.

(5) Please provide methods and further details about the blood draw to gather plasma samples. 

(6) Please remove the description of “fasted cortisol” in the abstract.  According to your methods, patients were not fasted.  Authors were simply following the kit instructions for no food, water or teeth brushing 30 min before saliva was collected.  

(7) Please correct the grammatical error in your statistical analysis method descriptions from two-tiled to two-tailed.

(8) In the Western blot analysis, typically your immune-signal is compared to the total protein of the entire Coomassie stained lane.  I suggest that authors provide an image of the entire gel, not just a select band of the stained gel.

(9) The Discussion would benefit from paragraphs to break up main ideas.

Author Response

Thank you for your advice and constructive comments concerning our manuscript ID: healthcare-1122883 entitled “Redox homeostasis and metabolic profile in young female basketball players during In-season training.” Authors: Rosamaria Militello, Simone Luti, Matteo Parri, Riccardo Marzocchini, Riccardo Soldaini, Alessandra Modesti, Pietro Amedeo Modesti. We have carefully considered reviewers’ suggestions, and have revised our manuscript accordingly; we hope that these changes meet with your approval. The changes are marked.

These are the comments and in italics our replay

Reviewer 1

Suggestions to authors:

(1) Treat your control, as your control or "normal" to make your results more readable. Accordingly, report your data in reference to your control not your experimental group; for example, "basketball players had 51% less XXX than control."  NOT, "control had 51% more XXX compared to basketball."

We agree with this comment and we modify as reported below:

3.2. Plasma oxidative stress measurements

“…..as shown in Figure 1A the BAP mean value was 1774.73 ± 169 µmol/L for basketball players in comparison with 1465.04 ± 139 µmol/L for controls. Moreover as shown in Figure 1B, the basketball players show a decrease of 21.9% (P < 0.05) of the levels of oxidative species (dROM), the mean value for athletes was 280.5 ± 38 UCarr and for control group was 359.75 ± 44 UCarr.”

3.4. Salivary cortisol determination

“The results, reported in detail in Figure 1D show a significant decrease (51%; P < 0.009) in basketball player in comparison with control group. In detail, the cortisol mean value was 915.09 ± 613.69 pg/ml for basketball players and 1867.68 ±659.83 pg/ml for controls.”

3.5. Plasma metabolic profile

” In plasma from athletes we found an increase in lactic acid (460%; P= 0,0013), urea (220%; P= 0,0361) and ornithine (130%; P= 0,0467) and a reduction in hydroxybutyric acid (75%; P= 0,0258), L-glutamic acid (83%; P= 0,0353) and L-asparagine (82%; P=0,0498).”

(2) Authors refer to the basketball player subjects as “professional” or “elite” athletes but don’t provide other information to further define or support these terms other than that they trained 5 times a week for 2h per session.  Please further define these parameters with regards to your introduction about chronic or acute training.  For example, how many months or years have they been doing this?

We agree and to better clarify we added this sentence in “2.1. Participants” paragraph:

“The elite-level female athletes were recruited from the local sports clubs in Florence Italy “US AFFRICO-Firenze” who have been practicing this sport for more than 5 years. …….. Team coaches followed a classical training program that we can resume as follow: half an hour of low-moderate run followed by interval training runs with sprinting and repeating to improve speed and simulate the different game situations. Then move on to strengthening the muscles involved in basketball: A) work on the shoulder muscles by performing the classic front and side lifts with weight lifting dumbbells: three sets of twelve repetitions. B) work with the abdominals (oblique and the rectus abdominis muscle) which are essential to maintain balance in every movement and to contribute to a good deadlift in jumps: three sets of 15 repetitions.”

(3) Grammar and spelling throughout the manuscript need to be addressed.

We agree, we checked the manuscript and we have made the appropriate corrections.

(4) Authors don’t give any information about how and where they recruited the Control group.  They also don’t give any details of what they are trying to “match” for the Control characteristics.  For example, are they doing an age-matched control group?  Ethnicity-matched control group?  BMI-matched control group?  Please explain.   Authors also don’t give any information regarding ethnicity of any of the participants.

We agree with this comment and we added this sentence in paragraph “2.1 Participants”:

“The control female group was recruited among students of the degree course in Motor Sciences, Sport and Health of the University of Florence. All female subjects volunteered to participate following explanation of all experimental procedures were adults (≥20 years) and of Caucasian ethnicity.”

To better specify we added this sentence in 3.1. Participant’s characteristics paragraph:

“The female control students were age/BMI matched with the athletes and descriptive characteristics of the participants are reported in Table 1 showing that mean age was 26 ± 4.13 years for all the volunteers (athletes and non-athletes)”

(5) Please provide methods and further details about the blood draw to gather plasma samples. 

We used a Safety-Lancet (Sarstedt AGand Co) against the cleansed puncture site; we gently massaged the finger in the direction of the puncture site to obtain the required blood volume. We collected it on heparinized microvette (Sarstedt). For plasma separation, we centrifuged the samples for five minutes at 2000 x g.   

To better specify the method we added this sentence in paragraph “2.1 Participants”:

“Capillary blood sample is taken using a heparinized Microvette CB300 (Sarstedt) from each volunteer after obtaining written and informed consent. Holding the Microvette in a horizontal position the blood sample was collected with the capillary tip (filling volume 300 µl).”

(6) Please remove the description of “fasted cortisol” in the abstract.  According to your methods, patients were not fasted.  Authors were simply following the kit instructions for no food, water or teeth brushing 30 min before saliva was collected.  

We agree, we removed it.

(7) Please correct the grammatical error in your statistical analysis method descriptions from two-tiled to two-tailed.

We agree, we corrected it.

(8) In the Western blot analysis, typically your immune-signal is compared to the total protein of the entire Coomassie stained lane.  I suggest that authors provide an image of the entire gel, not just a select band of the stained gel.

We agree, we provide an image of the entire gel and modify Figure 1C:

(9) The Discussion would benefit from paragraphs to break up main ideas.

We agree, and we divided chapter “4. Discussion” in:

4.1 Redox profile

4.2 Adiponectin

4.3 Salivary cortisol

4.4 Metabolomic profile

Moreover we added chapter “5. Conclusion”:

“Despite that the main limitation of our study is the low number of participants analyzed these data indicate that trained women have greater antioxidant capacity and a reduction in oxidative species than sedentary women, differences that are associated with an increase in plasma lactate level. These data complement recent observations suggesting the participation of lactate in the antioxidant potential through an intracellular buffering action that would allow it to neutralize or delay the creation of an acidic environment in muscle cells [Ref. 31:  Robergs RA, Ghiasvand F, Parker D. Biochemistry of exercise-induced metabolic acidosis. Am J Physiol Regul Integr Comp Physiol. 2004 Sep;287(3):R502-16. doi: 10.1152/ajpregu.00114.2004. PMID: 15308499].. The power of these actions seems to be enhanced with training in women. Moreover, lactate metabolism fulfills others functions: an energy source, a gluconeogenic precursor and a regulatory molecule a “lactormone” [Ref. 32: Brooks GA. Lactate as a fulcrum of metabolism. Redox Biol. 2020 Aug;35:101454. doi: 10.1016/j.redox.2020.101454. Epub 2020 Feb 9. PMID: 32113910; PMCID: PMC7284908.]. Lactate produced after a high cell work, is secreted into the interstitial and circulation and in diverse tissues it acts by cell redox regulation, ROS generation and lactylation of histones. Moreover regular exercise results in repeated lactate exposure leading the adaptive processes such as mitochondrial biogenesis and other healthful characteristics such as improved metabolic flexibility. For this reasons the fact that in our experimental conditions we find high levels of lactate, low levels of adiponectin as well as low salivary cortisol can lead to the conclusion that the type of training in basketball induces adaptation in well-trained athletes without causing inflammatory phenomena and avoids the onset of overtraining/overreaching.”

Reviewer 2 Report

This study confirmed the Redox homeostasis and metabolic profile of young female basketball players during training during the season.

Since most of the research on oxidative stress indicators and homeostasis has been done in male athletes, their value can be appreciated. It is believed that the quality of the manuscript will be higher if appropriate corrections are made.

1. What are the technical and aerobic exercises mentioned in the 2.1 participants section? Please be specific.

2. What does "All" mean in Table 1?

3. In Figure 1, you must additionally mention the sign of significant difference. Currently there is only about ****.

4. In the discussion section, The authors note that there is no research on adiponectin and female athletes. Are there any studies analyzing adiponectin in female athletes?

5. In the discussion section, the author states that the training program used was either low-intensity or adapted to the training program. So, in other studies, is it because athletes with high cortisol levels had high intensity in their training program or did not adjust well to the training program? We need more specific logic. In general, athletes perform their maximum workouts during the season.

6. As a result of metabolite analysis, female basketball players showed significantly higher lactate levels compared to the control group. As you know, lactic acid is a biomarker that evaluates exercise levels, so why is lactic acid high compared to low adiponectin? You need to explain why.

7. At the end of the discussion, It need the findings and suggestions from this study.

Author Response

Thank you for your advice and constructive comments concerning our manuscript ID: healthcare-1122883 entitled “Redox homeostasis and metabolic profile in young female basketball players during In-season training.” Authors: Rosamaria Militello, Simone Luti, Matteo Parri, Riccardo Marzocchini, Riccardo Soldaini, Alessandra Modesti, Pietro Amedeo Modesti. We have carefully considered reviewers’ suggestions, and have revised our manuscript accordingly; we hope that these changes meet with your approval. The changes are marked.

These are the comments and in italics our replay

Reviewer 2

Suggestions to authors:

  1. What are the technical and aerobic exercises mentioned in the 2.1 participants section? Please be specific.

As reported in the 2.1 paragraph the athletes trained 5 times a week and the training protocol involved technical and aerobic exercise. To better explain the protocol followed, we added this sentences to the “paragraph 2.1 Participants:

“Team coaches followed a classical training program that we can resume as follow: half an hour of low-moderate run followed by interval training runs with sprinting and repeating to improve speed and simulate the different game situations. Then move on to strengthening the muscles involved in basketball: A) work on the shoulder muscles by performing the classic front and side lifts with weight lifting dumbbells: three sets of twelve repetitions. B) work with the abdominals (oblique and the rectus abdominis muscle) which are essential to maintain balance in every movement and to contribute to a good deadlift in jumps: three sets of 15 repetitions”

  1. What does "All" mean in Table 1?

“All” means data from Basketball players + controls. To better clarify we substituted All with “all the study participants”

  1. In Figure 1, you must additionally mention the sign of significant difference. Currently there is only about ****.

We agree and we added the sign of significance for two asterisks.

  1. In the discussion section, The authors note that there is no research on adiponectin and female athletes. Are there any studies analyzing adiponectin in female athletes?

We are sorry but we do not understand what the reviewer means because in discussions we reported several study analyzing adiponectin in female athletes during exaustive or chronic exercise

[22.      Morrison AE, Fleming S, Levy MJ. A review of the pathophysiology of functional hypothalamic amenorrhoea in wo-men subject to psychological stress, disordered eating, excessive exercise or a combination of these factors. Clin Endo-crinol (Oxf). 2020 Dec 20. doi: 10.1111/cen.14399. Epub ahead of print. PMID: 33345352.

  1. Plinta R, Olszanecka-Glinianowicz M, Drosdzol-Cop A, Chudek J, Skrzypulec-Plinta V. The effect of three-month pre-season preparatory period and short-term exercise on plasma leptin, adiponectin, visfatin, and ghrelin levels in young female handball and basketball players. J Endocrinol Invest. 2012 Jun;35(6):595-601. doi: 10.3275/8014. Epub 2011 Oct 6. PMID: 21986458.
  2. Hornsby WG, Haff GG, Suarez DG, Ramsey MW, Triplett NT, Hardee JP, Stone ME, Stone MH. Alterations in Adiponec-tin, Leptin, Resistin, Testosterone, and Cortisol across Eleven Weeks of Training among Division One Collegiate Thro-wers: A Preliminary Study. J Funct Morphol Kinesiol. 2020 Jun 19;5(2):44. doi: 10.3390/jfmk5020044. PMID: 33467260; PMCID: PMC7739239.]

but we do not know studies reporting adiponectin levels in trained athletes compared with control women. 

However to better clarify we modify this sentence in discussion (4.2 Adiponectin):

“However, there are little data on the relationship between muscular fitness and adiponectin levels. We are not aware of any study on associations between adiponectin and training in female athletes.”

With this:

“However, there are little data on the relationship between muscular fitness and adiponectin levels as well as on associations between adiponectin and training in female athletes”

  1. In the discussion section, the author states that the training program used was either low-intensity or adapted to the training program. So, in other studies, is it because athletes with high cortisol levels had high intensity in their training program or did not adjust well to the training program? We need more specific logic. In general, athletes perform their maximum workouts during the season.

As reported by Taipale RS et al [ref 12 in manuscript] in women, cortisol level shows an increase in response to an acute bout of endurance exercise but it appears to decline with training. Moreover according to Consitt LA. [ref 13 in manuscript] an elevated resting cortisol level is associated with overtraining in athletes. In our experimental conditions, we can say that the athletes are trained with a program that avoids the onset of overtraining/overreaching as reported in chapter “5. Conclusion”

  1. As a result of metabolite analysis, female basketball players showed significantly higher lactate levels compared to the control group. As you know, lactic acid is a biomarker that evaluates exercise levels, so why is lactic acid high compared to low adiponectin? You need to explain why.

Thank you for this comment because it allows us to deepen the importance of lactate as a signal molecule. Lactate has always been considered as a metabolic waste product and fatigue agent but recently as reported by Brooks G.A. [Brooks GA. Lactate as a fulcrum of metabolism. Redox Biol. 2020 Aug;35:101454. doi: 10.1016/j.redox.2020.101454. Epub 2020 Feb 9. PMID: 32113910; PMCID: PMC7284908.] its metabolism fulfills others functions: an energy source, a gluconeogenic precursor and a regulatory molecule a “lactormone”.

  1. At the end of the discussion, It need the findings and suggestions from this study.

As suggested by Rev 1 “The Discussion would benefit from paragraphs to break up main ideas.” We divided chapter “4. Discussion” in:

4.1 Redox profile

4.2 Adiponectin

4.3 Salivary cortisol

4.4 Metabolomic profile

We added chapter 5. Conclusion:

“Despite that the main limitation of our study is the low number of participants analyzed these data indicate that trained women have greater antioxidant capacity and a reduction in oxidative species than sedentary women, differences that are associated with an increase in plasma lactate level. These data complement recent observations suggesting the participation of lactate in the antioxidant potential through an intracellular buffering action that would allow it to neutralize or delay the creation of an acidic environment in muscle cells [Ref. 31: Robergs RA, Ghiasvand F, Parker D. Biochemistry of exercise-induced metabolic acidosis. Am J Physiol Regul Integr Comp Physiol. 2004 Sep;287(3):R502-16. doi: 10.1152/ajpregu.00114.2004. PMID: 15308499.). The power of these actions seems to be enhanced with training in women. Moreover, lactate metabolism fulfills others functions: an energy source, a gluconeogenic precursor and a regulatory molecule a “lactormone” [Ref. 32: Brooks GA. Lactate as a fulcrum of metabolism. Redox Biol. 2020 Aug;35:101454. doi: 10.1016/j.redox.2020.101454. Epub 2020 Feb 9. PMID: 32113910; PMCID: PMC7284908.]. Lactate produced after a high cell work, is secreted into the interstitial and circulation and in diverse tissues it acts by cell redox regulation, ROS generation and lactylation of histones. Moreover regular exercise results in repeated lactate exposure leading the adaptive processes such as mitochondrial biogenesis and other healthful characteristics such as improved metabolic flexibility. For this reasons the fact that in our experimental conditions we find high levels of lactate, low levels of adiponectin as well as low salivary cortisol can lead to the conclusion that the type of training in basketball induces adaptation in well-trained athletes without causing inflammatory phenomena and avoids the onset of overtraining/overreaching.”

Round 2

Reviewer 2 Report

The author has sincerely answered the reviewer's questions and is well reflected in the script.